# Artsheets for Art Datasets

**Ramya Srinivasan**
Fujitsu Research of America

**Emily Denton**
Google Research, New York

**Jordan Famularo**
University of California, Berkeley

**Negar Rostamzadeh**
Google Research, Montreal

**Fernando Diaz**
Google Research, Montreal

**Beth Coleman**
University of Toronto

## Abstract

Machine learning (ML) techniques are increasingly being employed within a variety of creative domains. For example, ML tools are being used to analyze the authenticity of artworks, to simulate artistic styles, and to augment human creative processes. While this progress has opened up new creative avenues, it has also created the opportunity for adverse downstream effects such as cultural appropriation (e.g., cultural misrepresentation, offense, and undervaluing) and representational harm. Many of the concerning issues stem from the training data in ways that diligent evaluation can uncover, prevent, and mitigate. As such, when developing an arts-based dataset, it is essential to consider the social factors that influenced the process of conception and design, and the resulting gaps must be examined in order to maximize understanding of the dataset's meaning and future impact. Each dataset creator's decision produces opportunities, but also omissions. Each choice, moreover, builds on preexisting histories of the data's formation and handling across time by prior actors including, but not limited to, art collectors, galleries, libraries, archives, museums, and digital repositories. To illuminate the aforementioned aspects, we provide a checklist of questions customized for use with art datasets in order to help guide assessment of the ways that dataset design may either perpetuate or shift exclusions found in repositories of art data. The checklist is organized to address the dataset creator's motivation together with dataset provenance, composition, collection, pre-processing, cleaning, labeling, use (including data generation), distribution, and maintenance. Two case studies exemplify the value and application of our questionnaire.

## 1 Introduction

Machine learning (ML) technologies are rapidly being integrated into creative arts domains in order to analyze artistic content [1, 2], support existing artistic practice [3, 4], and establish novel methods of creating and re-mixing media [5, 6]. Because of the crucial role that the arts play in society, through culture [7], discourse [8], and the economy [9], understanding the role of ML in the broader art world is critical to responsible research and development of ML for arts.

Although motivated by supporting new artistic practices, ML-assisted or ML-based digital art and associated technologies bring with them ethical concerns ranging from representational harm to cultural appropriation [10, 11, 12, 13]. Many of the ethical concerns with ML technologies in creative domains can be traced back to the datasets underlying their development. The recognition of data as a key factor contributing to downstream ethical concerns builds upon a growing body of scholarship

35th Conference on Neural Information Processing Systems (NeurIPS 2021) Track on Datasets and Benchmarks.

focused around ML datasets and practices of dataset development. For example, scholars have identified a widespread devaluation of slow and careful dataset development [14], where speed and scale are often prioritized at the expense of care for data subjects, data creators, and data laborers [15]. Studies have shown that incomplete or ambiguous label guidelines can result in differing interpretations of concepts and inconsistent labels [16]. Dataset audits have uncovered a host of concerns with the ways minoritized groups are represented – or fail to be represented at all – in prominent ML datasets [17, 18, 19, 20]. Unstandardized dataset documentation,and maintenance frameworks have been identified as a key barrier to reproducibility and accountability [21, 22, 23, 15].

In response to the growing ethical challenges in dataset development, a suite of frameworks and guidelines have been proposed to mitigate the aforementioned concerns. For example, [24, 25] draw lessons from the standardized data practices within cultural heritage studies and call for a similar professionalization of data work within machine learning. In [26], a data annotation method that aims to warn about the risk of discriminatory results of a given dataset is proposed. Documentation frameworks have been developed to facilitate transparent dataset reporting and promote accountable dataset development such as [21, 27, 23], and [28]. While these are effective generic approaches to dataset development, they risk losing important nuance specific to a domain, such as the arts.

In this work, we extend and complement previous dataset documentation efforts, focusing specifically on the unique considerations (such as social, legal, cultural, historical, and environmental factors) that arise in the development of art datasets. In so doing, we counter a common presumption of "one-size-fits-all" ethics checklists in the field by reinforcing the principle that ethical frameworks must be carefully adapted to each use and providing a roadmap for implementing it. Specifically, we augment the Datasheets questionnaire [21], and tailor it towards art specific datasets, addressing areas that are specific to arts and culture. Our checklist, which we term *Artsheets*, is designed to offer dataset creators with a framework to facilitate critical reflection of various aspects of their work and offer guardrails for types of arts-based datasets now in popular use by the ML community [29, 30, 31, 32, 33, 34]. Additionally, we anticipate Artsheets having relevance to a broad set of stakeholders beyond ML dataset creators, serving as a boundary object that will create new affordances for communication and decision making that crosses community boundaries. For example, we anticipate Artsheets will support dataset consumers in making informed decisions relating to appropriate dataset use. Artsheets may also support policy decisions relating to the digitization of artefacts from cultural heritage institutions and offer artists with information necessary to maintain agency over the distribution of their creations. Finally, our framework can also be leveraged as a tool for retrospective and historical dataset analysis and documentation.

The remainder of this paper is structured as follows. First, we outline the process by which we developed Artsheets. Next we detail the Artsheets questions and workflow. To facilitate easy comparison with Datasheets [21], we make note of questions that have been adapted directly rather that being wholly novel to our framework. Next, we summarize our findings from two case studies of Artsheets applied retrospectively to document ArtEmis [32], a visual art dataset, and JS Fake Chorales [35], a generated music dataset. Our full case studies can be found in the appendix. Finally, we close by discussing limitations, challenges to implementation, and future work.

## 2   Development Process

We conducted a three month project to develop a questionnaire for art datasets. We adopt an interdisciplinary approach to our development process, consistent with current practices in the field [36, 37]. Our questionnaire incorporates perspectives from experts across computer vision, ML, law, HCI, ethics, generative modeling, and music.

We started with the set of questions provided in [21] to examine the feasibility and sufficiency of these questions in the context of art datasets. Drawing on our experiences as researchers with diverse backgrounds (art history, ML, computer vision, ethics, HCI, and science and technology studies) working in different institutions (industry and academia), we leveraged our knowledge on ML-based art applications, dataset curation, and associated ethical issues to identify a set of questions specific to art datasets, focusing on visual arts and music datasets. In the process of developing the questions, we also consulted existing curatorial guidelines and best practices such as [38, 39]. We then evaluated the applicability of the developed questions on a visual art dataset [32], modifying and refining the questions based on the evaluation exercise. In particular we reformulated provenance and

collection issues from arts and archaeology fields to suit ML contexts, added new considerations for data generation due to its prevalence and associated risks, and re-organized the contents of sections, merging common aspects. We then evaluated the questions on a generated music dataset [35]. In parallel, we also shared the set of questions with colleagues and experts across diverse fields and subsequently incorporated their feedback to refine the questions further.

## 3 Artsheets Questions and Workflow

The following sub-sections represent an extension of the Datasheets [21] questionnaire, tailored towards art datasets. While some questions are similar to the original Datasheets questions, many others are unique and specific to art datasets. In particular, we include two new question categories, data provenance and data generation, that become relevant in the context of art datasets. Several art-specific questions concerning dataset creator motivation, data composition, collection, pre-processing, cleaning, labeling, use, distribution, and maintenance are included in the corresponding sub-sections. Questions that have been adapted from the original Datasheets questions are marked by a dagger (†), and those questions without a dagger are new additions. Each sub-section begins with background information to substantiate the need for the questions contained therein. To exemplify the practical use of our questionnaire, in the appendix, we provide two full case studies, illustrating the use of Artsheets on two recent ML datasets, namely, ArtEmis [32], a visual art dataset and JS Fake Chorales [35], a generated music dataset. The selection was determined in part because curation of these datasets is sufficiently publicly documented to allow analysis. Section 4 summarizes the main takeaways from these case studies.

### 3.1 Motivation

The motivation section of Datasheets focuses primarily on the motivations underlying development of the ML dataset in question. Here, we extend the original questions based on the observation that the development of many art datasets rests on multiple layers of curation, each with unique motivations. For example, many art datasets are drawn from digitized collections of art, archaeological finds, and other artifacts drawn from human history. The physical collections themselves have been initiated, maintained, and modified by complex series of individual and institutional imperatives. Acquisition decisions made by collectors, curators, librarians, and institutional boards of directors determine which available pieces are included or not included, and under which conditions such as price or lending agreements. Although cultural inclusiveness has become an essential curatorial responsibility, particularly for large encyclopedic cultural institutions, studies have shown that the primary cultural narrative reflected in many art collections is grounded in principles of exclusion [40]. For example, a recent study showed that leading audio streaming platforms like Spotify mediate markets to advance their own interests [41]. A separate study concerning artist diversity across 18 major U.S. museums representing the work of more than 9000 artists showed that $85\%$ of them were white and $87\%$ were men [42]. Second, artists with a strong peer network are more likely to succeed in getting their works showcased. Studies have shown that artists connected with prestigious artists through "strong symbolic ties" (i.e., repeated exhibitions) tend to garner the most historical recognition [43, 44]. As a result, works of emerging, lesser known, and underappreciated artists may be sidelined and under-represented in the resulting datasets. Third, not all GLAMs (galleries, libraries, archives, and museums) are equal: small organizations do not have the funding to use technology at the same level as the large ones. This circumstance affects which artworks are digitized and made available online [45].

In short, the choice of underlying data source is highly consequential as new curatorial decisions layer upon previous ones. The following questions pertain to the motivations of art dataset development that relate both to dataset purpose and underlying choice of data source.

1. Who funded the curation of the ML dataset, and for what purpose? Who was involved in the curation process? Please list all individuals and/or institutions.†

2. If the ML dataset was sourced from an underlying curated source (e.g. a museum), please outline the factors governing the inclusion of artworks into the source collection.

3. If the ML dataset was sourced from an underlying curated source, please outline the factors motivating the choice of the data source.

## 3.2 Data Provenance

Provenance traces the "biographies" of artworks from the time they were created—their whereabouts in time and space [46]. Provenance studies are extremely important for art datasets, not just for legal considerations such as ownership, title, and authenticity, but also for understanding the broader social, historical, and cultural contexts pertaining to appreciation, attribution, display, acquisition, and value of the artworks [38]. Be it for verifying subjects in portraits [47] or for estimating the value of artworks [48], machine learning tools are employed for a diverse range of art-related tasks. Provenance research becomes even more important in such applications. For example, the author in [49] discusses the changing provenance of a 17th-century painting by Carel Fabritius citing instances of misidentification, misattribution, and lost documentation. An ML application that overlooks the evolving character of provenance information can potentially misidentify an artwork and distort its history, and thereby amplify misunderstanding. In addition, the scale of ML-generated art could amplify the harm of not identifying the origins of the artwork, and make it more difficult to trace back this information as a variety of art styles may be blended into a single narrative without acknowledging the source and background of the datasets. Lack of provenance information in ML applications can also result in misattribution (e.g. by crediting an artwork to an artist not associated with the artwork), or even in undervaluation of artworks (e.g. by predicting low market prices). While it may be challenging for the ML community to obtain accurate information concerning data provenance, including domain experts such as art historians, art connoisseurs, museum curators, individual buyers and lenders, artists, auction houses, and other relevant stakeholders can help overcome some of these challenges and better inform the process of dataset curation. The following questions are not only applicable in the context of creating new art datasets but also for analyzing problem feasibility and downstream impacts.

1. Is there a provenance policy outlining the ethics and conditions of transfer of works in the collection? For example, were the visual artworks or music pieces obtained by means of transfer from other sources? Please describe the terms and conditions of provenance policy.

2. Does the dataset include ownership histories, condition of the artworks at the time of procurement/inclusion into the dataset, and legal documentation related to the transfers? Please describe any available information related to ownership histories.

3. What type of information about the music piece/visual artwork's attribution and origination is included? Is it limited to a single creator, or does the dataset afford other categories such as the name of the person who commissioned the artwork, workshop assistants, symphony performers' names, its owners or stewards over time, names of remixers or transliterators, etc.? Please list any available information concerning the people associated with the artwork.

4. From which source(s) does the dataset draw authenticity verification and title of artworks and music pieces? Please name specific cultural institutions, private or public repositories, and all other sources. Please describe the types of personnel, organizations, and processes involved in the production of authentication and title information.

5. Is information pertaining to the music piece/visual artwork's appreciation included in the dataset? For example, information about multiple interpretations of an artwork/music piece, reviews, labels imprinted on the artworks/music pieces, etc. Please describe.

6. Are the values of Indigenous and minoritized groups honored while including the visual artwork/music piece in the dataset? Describe the process or protocol by which visual artworks/music pieces affiliated with these groups have been approved for use in the dataset.

## 3.3 Data Composition

To make informed decisions about how to use datasets, consumers need information about data composition, which is also needed by those who evaluate datasets for scholarly rigor and scientific value. Special issues arise in the composition of art datasets because they cannot be considered as random samples of the larger body of work they represent.

Art datasets are non-random and not necessarily representative samples of larger sets such as a cultural institution's total holdings or of the entire historical corpus of art. Individual instances are either present in or absent from the dataset due to many cumulative factors such as individual decisions, societal conditions, and environmental forces.

Though the impact of past events on data composition is complex and difficult to generalize, a comprehensive assessment of a specific project will likely uncover important nuance. Researchers ought to note that physical degradation affects data composition because it gradually or suddenly extinguishes artworks and music recording materials due to interactions between their tangible stuff and the conditions in which they are kept. Data composition is also sociopolitical: people have made context-dependent choices about which objects to keep versus discard or recycle. People and organizations in places of power are generally afforded more opportunities to collect and maintain works of art, and to impose their preferences. Dataset creators must be aware that, from antiquity to the present, the activity of collecting has often been inextricable from projects that are imperialist, oppressive, or fractious, and which reinforce the position of so-called superior groups that extract art and artefacts from marginalized people [50, 51, 52, 53, 54]. Data composition may also reflect professional norms: art collections are results of individual choices made in institutional settings about which objects to collect versus forego, and at what price and effort, according to conventions and practices that belong to occupations in the art-collecting sector and GLAMs. Decisions may be grounded in policies and processes established at micro- and macro-levels such as auction houses, the boardrooms of nonprofit organizations, codes of ethics for museum professionals, and national or international governance frameworks. Occupational norms shape how collections of artworks are digitized and made ready for dataset development by converting their original media into form suitable for computation.

When curating a dataset, it is essential to ask what is missing and why. We build on critiques of universal views of data collection and analysis derived from technoscience scholars and critical race theory practitioners [55, 56]. In other words, the work of Artsheets for art data sets is to help the ML community identify the partial view or "situated knowledge" of a dataset. The following questions are intended to guide such an inquiry.

1. What real-world objects or phenomena do the data instances represent (e.g., material objects, performances)?[†]

2. If the dataset has been derived from a larger curated corpus, please describe the relationship between the dataset and the larger sample by explaining the conditions (e.g., environmental, sociopolitical, and professional) that affect the relationship between the dataset and total corpus.

3. How many instances are there in total, including number of instances per type (e.g., number of artworks/music pieces per artist )? Are there recommended data splits (e.g., training, development/validation, testing) based on the number of instances per type? Please provide a description of these splits, explaining the rationale behind them.[†]

4. Is the dataset limited by intellectual property restrictions imposed by an external resource (e.g. a cultural institution's decision to exclude works not in the public domain)? Please provide descriptions of external resources and any associated restrictions, including links to access points and policies as relevant.

5. If data instances are proxies for visual objects, is there information about how the objects were converted to images? E.g., type of camera/model, camera parameters, lighting conditions, etc.? And is there a justification for the choice of the specific viewpoint, camera, and lighting conditions? What aspects of the artwork/object may be missing as a consequence of the specific viewpoint/lighting conditions? Please describe.[†]

6. If data instances are proxies for musical pieces, is information available about how performances were recorded? E.g. type of equipment, studio or live audience setting? Which aspects of the performance may be missing as a consequence of recording conditions? Please describe.[†]

7. If data instances are proxies for visual objects, which view(s) of the original objects do data instances represent? E.g., top view, side view, etc.?

8. Does the dataset contain opinions, sentiments, or beliefs that relate to specific social or cultural groups (e.g., religious beliefs, political opinions, etc.) or visualizations/musical renderings that might cause anxiety (e.g., those depicting wars, oppression, etc.) ? If so, please describe.[†]

## 3.4 Data Collection

Several unique culture-specific factors and legal issues can arise while collecting art datasets, that the ML community should examine. First, the ML community should consider their responsibility to provide compelling experiences of art, culture, and creativity by providing wide ranging and inclusive perspectives [38]. Curated datasets should therefore ensure cultural relevance, inclusion, and longevity. Second, issues such as IP rights and transfer policies should be carefully studied in order to ensure that the dataset does not violates legal terms and conditions [57, 58]. Such violations can have serious legal consequences such as court trials and reparation. For example, "Pathur Nataraja," a 12th century bronze idol of Lord Shiva illegally sold to the Canadian Bumper Corporation, was repatriated to the Indian government after a case trial in Britain [59]. Third, and related to legal aspects is the issue of misappropriation of cultural artifacts, often in relation to Indigenous and post-colonial nations [60]. Repatriation of artworks pillaged during war or colonization has emerged as a hotly debated topic and one that may motivate deaccessioning of works in institutional collections [38]. For example, in 2018, French President Emmanuel Macron commissioned a report that recommended repatriation of objects extracted from Africa during the period of French colonization [61]. In the context of these issues, the ML community should examine the answers to the following questions concerning the data collection process.

1. How were the individual visual artworks or music pieces collected? Were they obtained directly from museums, galleries, music labels, archives, etc.? Or were they derived from existing ML datasets? Please provide details concerning the data acquisition process.[†]

2. Is there any institutional (e.g., museum, music label, etc.) policy governing the data collection process? Please describe any applicable institutional ethical standards/rules followed in collecting the data.[†]

3. Are all relevant stakeholders (e.g. artists, auction houses, galleries, museums, music labels, composers, legal experts, archivists, etc.) consulted in the collection process? For example, how has the dataset collection process been informed by art conservationists, art historians, and other stakeholders (e.g., Indigenous groups) in order to ensure cultural relevance and longevity? Please describe.

4. Is there information on the IP rights and cost of acquisition of an artwork or a music piece? Is there information pertaining to their usage, display, reuse, sale, reproduction, and transformation? Please describe all relevant terms and conditions.

5. Is there permission to collect/view/reproduce culturally sensitive information? For e.g., those pertaining to Indigenous and minoritized groups, secret objects, sacred objects, etc. that may be forbidden from public view? Please describe the details related to consent request and approval from the concerned individuals and/or communities.

6. Do the included works provide wide ranging and multi-dimensional narratives? Please describe how diversity and inclusion has been ensured in the data collection process—for e.g., demonstrate that the collection is not skewed by providing the number of works across different artists, sources, communities, and geographies; describe how the collection upholds ethical values of the groups it includes and showcases diverse perspectives.

## 3.5 Data Pre-processing, cleaning, and labeling

"Raw data" instances often undergo a number of transformations before being consumable by a model. Transformation includes removing information from individual instances using mechanical or manual operations. In terms of images, this might involve resizing, recoloring, or down-sampling. In terms of music, this might involve truncating a song, changing the sample rate, or converting audio into a score. Transformation can also include adding information from individual instances using mechanical or manual operations. In terms of images, this might involve segmentation, up-sampling, manually assigning categorical labels, or joining with side information. In terms of music, this might involve source separation, rhythmic detection, or joining with cultural data such as reviews.

Each decision to remove or add information can carry with it the values of the dataset designer. Removing information can literally erase subtle attributes of an artwork, and so preserving the original data is important. But, more importantly, the decision to erase some aspects and not others can be arbitrarily guided by the values of the dataset creator. Converting non-Western music to

Western notation is a clear example of this. Moreover, appending new data, be it manual or automatic, to an instance can carry profound implications for artists [62]. The manual process of labeling itself, common in ML dataset development pipelines, carries values in both the definition of categories and in the annotators assigning labels [63, 64]. The following questions are therefore included to guide the dataset annotation process.

1. Were there any specific pre-processing steps (e.g. filtering, resizing, rotations, etc.) that might have resulted in loss of information? Please describe.[†]

2. Were there any translations involved in obtaining the relevant metadata? e.g. translating description of an aboriginal artwork from a native language to English? If so, please describe the diligence process to ensure accurate translations.

3. Does the dataset identify any subpopulation (e.g. by ethnicity, gender expression) by virtue of text inscribed on the artwork or its frame? If so, describe whether such terms are captured in the annotation process and the method of transcription. Describe how such terms are interpreted in their original historical context and how the dataset's users and reviewers can trace the terms.[†]

4. Does the dataset identify any subpopulation in languages other than modern English (e.g. ancient Greek, modern French, modern creole and pidgin languages) or scripts other than standard English characters (e.g., types of Arabic script)? Describe how the dataset documents subpopulations by language(s) and script(s).[†]

5. Is there a policy/guideline related to the way labeling/annotation was done? For e.g., any (minimum and maximum) limit on the number of words/characters to describe the artwork, the relevant contexts (cultural, historical, social, etc.) the annotation should cover. Any specific protocol to label based on the type of artwork? Please describe. If the labels were already obtained from existing GLAMs, is there documentation pertaining to how the labels were created? For example, were there any institution specific guidelines? Please describe.

6. What information is included in the labels? For example, artist's name, artist's nationality, date associated with the artwork, associated art movement (Byzantine, Renaissance, etc.), genre (classical music, portraits, etc.), art material (oil, paint, etc.), source of the artwork (museum name, name of the recording label, etc.), etc. Please describe.[†]

7. Is there sufficient provenance information (e.g. historical, political, religious, cultural evidence, artist, date, art period, etc.) pertaining to the artwork to generate reliable labels? If not, describe how the labels were obtained/created?

8. Describe the background of the annotators (e.g., education, geographic location, their socio-cultural contexts, etc.) Do they possess relevant qualifications/knowledge to label the instance? If so, please justify. Are there some aspects of their specific backgrounds (e.g. political affiliations, cultural contexts, etc.) that could have biased the annotations? If so, please describe.

## 3.6 Uses and distribution

Although digitization of artworks can offer many benefits [45], it also invites conflicts, particularly at the intersection of copyright law with cultural identity and cultural appropriation [57, 65]. Cultural appropriation can occur when dominant cultures adopt elements from a non-dominant culture in a manner that misrepresents, or entirely loses, the original context and meaning. Some of the harmful impacts of cultural appropriation include devaluation of cultural heritage, cultural misrepresentation, and cultural offence [66]. For example, cultural artifacts often serve as an inspiration to fashion designers, who may modify the aesthetics associated with the original artifacts to propose new fashion styles (e.g., clothing, hairstyles, facial makeups, etc.). In doing so, the cultural significance, religious sentiments, and other sensitive values of local communities (from whom the fashion pieces were inspired) may be disrespected and compromised [67]. Any ML dataset that blindly borrows such fashion styles serves as a catalyst of cultural misrepresentation. Furthermore, existence of a power imbalance between the cultural appropriator (e.g. dataset creators) and those whose cultural values are appropriated (e.g. Indigenous groups), the absence of consent, and the presence of profit that accrues to the appropriator only amplify issues related to cultural appropriation [68, 69, 70]. The ML community should be aware of the cultural nuance associated with the original artworks and music pieces, and must ensure that in using and distributing the dataset, all aspects related to human

rights and dignity are upheld. The following questions are intended to guide the ML community in responsible use and distribution of art datasets.

1. What are the terms and conditions pertaining to the use and distribution of the dataset? For example, are there any licenses, required clearances from ministries of culture? Please describe the terms and conditions related to such license/clearance.[†]

2. Is there a link or repository that lists all papers/works that have used the dataset? What are some potential applications and scenarios where the dataset could be used?[†]

3. If the data concerns people (such as Indigenous groups), have they provided consent for the use and distribution of the dataset in the context of the specific application? If the dataset is going to be distributed, please describe the diligence process for ensuring that there are no existing restrictions on sharing the artworks and metadata.[†]

4. Are there any aspects of human rights and dignity that are prone to be suppressed in a specific use or distribution of the dataset? Please describe applications and scenarios where the dataset should not be used and/or distributed.[†]

5. If the dataset will be distributed under a copyright or other intellectual property license, and/or under applicable terms of use, please describe whose (individual or group's) interest/stake in the data is likely to be promoted, and whose interest/stake is likely to be suppressed, and why so. Please justify the use/distribution of the dataset given this context.

## 3.7 Data Generation

The rapid progression of ML technologies, and more specifically generative models, has contributed to the creation of a new set of algorithmic tools for artists. From mimicking artists' styles to creating new artworks, ML technologies are employed to generate music [71, 72], visual arts [73, 74], writing [75], and several other art forms [76]. While this progress has opened up new avenues, it has also raised ethical concerns such as biases and stereotypes embedded in generative art, as well as environmental impacts of generative art [12, 13, 11]. Moreover, as creating artistic contents through generative models require using existing data as training examples, generative art can raise questions related to ownership, credit assignments, and origins of the generated piece. Through the following questions, we cover some of the aforementioned issues surrounding datasets that contain generated contents and those concerning the use of samples from real datasets to generate new artworks.

1. Is there any content in this dataset that is generated by ML models? If so, please explain the source data that is used as training examples to generate the desired output.

2. If there are ML-generated contents in the dataset, are there any data licences required for using or distributing the training content?

3. Are there any permissions and data licences that dataset consumers need to adhere to in order to use the samples from this dataset to create any synthetic data (i.e: generated data using ML model)?

4. If there are generated contents in the dataset, is there documentation describing the computational costs involved in creating the data? Please describe

5. What could be some potential biases associated with the generated data? Please describe

## 3.8 Data Maintenance

The ML community should recognize the fact that art datasets are dynamic and evolving bodies of knowledge that need to be constantly updated. First, art datasets have to be updated to reflect the most accurate, and latest information concerning individual data instances. For example, there might be debates related to authenticity of some artworks. In such scenarios, dataset has to be updated to record the most up-to-date and accurate information. Second, if any work was removed from the original source (e.g., de-accessioned from a museum repository due to legal or other such issues), then the corresponding data instance should also be removed from the ML dataset. Third, ML scientists/ developers should re-examine the datasets on a regular basis to ensure that the dataset is wide-ranging and inclusive. Dataset developers should constantly strive to incorporate diverse and evolving narratives into the dataset to reflect multiple viewpoints. For example, if there are new discoveries that have uncovered unknown past works, or if humanities research has shed new light on

existing notions about a visual artwork or music piece, the dataset should ideally be updated to reflect these aspects. Additionally, if there is a provision for the public to contribute to an existing art dataset, special care has to be taken to ensure that the information associated with the contributed instance is accurate, up-to-date, and that it upholds all aspects of human rights and dignity. The dataset owners should ensure that the contributed instances broadens narratives by supporting diversity and inclusion. The following questions are included to address the aforementioned points.

1. Who is maintaining the dataset and how can they be contacted?[†]

2. Is there a policy outlining the dataset update process? For example, will the dataset be updated as and when there is a change in information with respect to a data instance or will the dataset be updated at regular time intervals to reflect all changes in that time interval? Is there a description of the specific changes made- e.g., provenance related, reason for removal of a data instance, or addition of a data instance, etc. Please describe.

3. Is there a provision for others to contribute to the dataset? If yes, describe the process for ensuring the authenticity of the information associated with the contributed data instances.

## 4 Findings from the Case Studies

We chose ArtEmis [32] and JF Fake Chorales [35] datasets for retrospective analysis given the public availability of supporting documentation. The case studies shed light on important shortcomings concerning the curation process of the analyzed datasets.

First, both the datasets are largely Western-centric datasets, not necessarily showcasing diverse perspectives. Furthermore, cultural stakeholders such as art historians and art conservationists are not consulted during the data collection process, thereby raising questions related to cultural relevance and longevity. In our assessment of data composition (outlined in Appendix A.3), we found that the ArtEmis dataset does not reach beyond WikiArt for sources. Hence, it reiterates the environmental, sociopolitical, and professional factors that shaped WikiArt as of 2015. Both WikiArt and ArtEmis necessarily exclude artworks that were destroyed or lost before the invention of photography. The datasets focus on a limited number of artistic media, styles, historical periods, and geographies. A high-level review suggests that their composition centers on modern Western artistic traditions and preferences, prioritizing certain media and styles from the six most recent centuries and only two continents (North America and Europe). The precise reasons for this reductive result are difficult to document due to the low availability of public information regarding the historical architecture of WikiArt. Generally, a series of choices made by its development team and external contributors determined which artworks are included versus not. In turn, such choices were constrained by prior factors that affected the composition of art repositories, such as those of GLAMs from which WikiArt derived its data. Our assessment of data composition for JS Bach Chorales (outlined in Appendix B.3) found that the training data was sourced from Music 21 toolkit, which in turn has been contributed by individual owners and institutions, the majority of which represent non-Western traditions. The rationale behind imitating JS Bach Chorales is not clear; we speculate that prior work in the ML community (that focused on imitating JS Bach chorales) could be one reason. This observation suggest that it is important for the broad ML community (beyond the arts domain) to recognize the intentional and unintentional limits of a dataset and calls for increase in the development of specifically curated datasets that are centered around non-Western perspectives.

Second, evaluation of questions related to data labeling (Section 3.5) showed that there is little to no information related to the identity or background of the annotators. For example, data instances in ArtEmis are accompanied by annotations reflecting subjective judgments of the artworks in question, including affective responses and textual explanations of the affective responses. ArtEmis dataset creators recognize the subjective nature of these judgments, and indeed characterize the subjective nature of the judgments as a feature of the dataset. However, no information about the social identities of the annotators is provided in the accompanying dataset documentation, thus it is difficult to determine whose perspectives are included in the dataset.

A third aspect that is quite common across most ML datasets (and therefore not necessarily specific to these datasets) is that there is very little provenance information associated with the datasets, making it difficult to verify the authenticity of the included information. In response to questions contained in Section 3.2 (and outlined in Appendix A.2, B.2), we found that the ArtEmis and JS Fake Chorales

dataset only provide minimal details pertaining to ownership histories and authenticity. Further, we found that Indigenous artifacts are underrepresented in these datasets. ArtEmis represents a limited selection of art, centered primarily on North American and European fine arts, which excludes other traditions and communities of practice, according to our review of the "art-styles" and "genres" that the ArtEmis creators have incorporated into the filtering tools for dataset exploration (`https://www.artemisdataset.org/#dataset`). The JS Fake chorales dataset represents synthetic symbolic polyphonic music samples imitating the style of German music composer JS Bach, and does not showcase any non-Western or Indigenous music forms. To address provenance issues, we believe including domain experts and cultural stakeholders (such as art historians, art conservationists, artists, auction houses, etc.) in the data curation process can be beneficial.

## 5    Limitations

Our discussion of Artsheets reflects the backgrounds of the authors. While expertise and feedback of people across diverse domains was leveraged in creating this checklist, it does not reflect the opinions of all stakeholders. For example, while we were able to involve art historians and consult generative artists, we were not able to directly work with traditional artists and policy makers. Hence, not all questions contained in our checklist are necessarily applicable across all forms of visual arts and music datasets, nor, despite our efforts, do they reflect a complete perspective of the visual arts. We consider Artsheets as a starting point that may be adapted in the future for other arts-based datasets, including fashion, dance, poetry, and narrative fiction and encourage practitioners to actively generate additional questions depending on their domain. Second, we have focused predominantly on datasets derived from human-made art; we have not covered all aspects of art datasets derived from machine generated content. Given that generative models find place in a variety of applications in the arts and beyond, it may be beneficial to define a set of specialized checklists and model cards such as [77] for generative models. Third, although we have considered dataset dynamics in the context of semi-automated manual maintenance, fully-automated *algorithmic* curation–such as techniques found in interactive learning–require scaling checklists across every algorithmic decision.

## 6    Conclusions

We have presented Artsheets as a contextualized variant of Datasheets for domains related to creative arts practice. As with more general Datasheets, we adopt a questionnaire-based approach to eliciting dataset information. Developing a questionnaire for art datasets required extending the generic questionnaire to include considerations of risks and harms unique to the art domain. Finally, through case studies, we demonstrate several categories of risks present in two art datasets. We emphasize that a customized, case-by-case analysis is required to explore the extent of assurance that dataset creators can offer regarding ethical use of Artsheets. It is to be noted that a variety of cultural stakeholders and domain experts like art historians, museum curators, archivists, artists, cultural policy makers, and lawyers need to be consulted in completing the Artsheets. In the long term, it may be therefore necessary for organizations and team composition to change significantly to address these issues adequately, for example, by means of including ethics specialists as full-time members of the organization. Given the importance of art more broadly in society, we believe that rigorous documentation of art datasets is necessary to understand and anticipate the broader implications of ML technologies. In future work, we intend to broaden the scope of analysis (including case studies in other artistic domains) and discourse (including other perspectives in the development of questions).

## Acknowledgements

We thank Geoff Anderson, Abhishek Gupta, Anna Huang, Hiroya Inakoshi, Devi Parikh, and James Wexler for their valuable feedback.

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

Here, we provide Artsheets for the ArtEmis dataset [32] and JS Fake Chorales dataset [35] mentioned in the main paper.

# A  Case Study: Artsheets for ArtEmis dataset

## A.1  Motivation

**Who funded the curation of the ML dataset, and for what purpose? Who was involved in the curation process? Please list all individuals and/or institutions.**

The dataset is authored by 5 researchers distributed across 3 institutions: Panos Achlioptas (Stanford University), Maks Ovsjanikov (LIX, Ecole Polytechnique, IP Paris), Kilichbek Haydarov (King Abdullah University of Science and Technology), Mohamed Elhoseiny (Stanford University; King Abdullah University of Science and Technology), Leonidas Guibas (Stanford University).

The work was funding by a Vannevar Bush Faculty Fellowship, a KAUST BAS/1/1685-01-01, a CRG-2017-3426, the ERC Starting Grant No. 758800 (EXPROTEA) and the ANR AI Chair AIGRETTE, and gifts from the Adobe, Amazon AWS, Autodesk, and Snap corporations.

The authors describe the dataset as the first attempt to associate visual art with human affective judgements and natural language explanations of the rationale behind each affective judgement. They motivate the development of this dataset by identifying "the formation of linguistic affective explanations grounded on visual stimuli" as an under-explored problem in computer vision.

**If the ML dataset was sourced from an underlying curated source (e.g. a museum), please outline the factors governing the inclusion of artworks into the source collection.**

All the images were sourced from the Wikiart dataset [78]. The WikiArt website states that the aim is to "make world's art accessible to anyone and anywhere." WikiArt is a community project – anyone can add or edit the content. We were unable to locate details about who the primary contributors are, e.g. geographical spread of contributors. However, we do know that at the time ArtEmis creators downloaded WikiArt images (2015), it consisted of fine art paintings from the 15th century until the 21st century, covering 1119 artists, 27 art-styles and 45 genres. Additionally, WikiArt states that the repository includes artworks are sourced from various civic buildings (e.g. museums, universities) as well as private collections.

**If the ML dataset was sourced from an underlying curated source, please outline the factors motivating the choice of the data source.**

The authors do not explicitly mention the factors behind choosing the Wikiart dataset. However, they do state that paintings and artistic photographs were considered for the analysis as they are prime examples of imagery created to elicit emotional responses from its viewers. Thus, we speculate that WikiArt was chosen because it is one of the largest publicly available collections of visual artworks.

## A.2  Data Provenance

**Is there a provenance policy outlining the ethics and conditions of transfer of works in the collection? For example, were the visual artworks or music pieces obtained by means of transfer from other sources? Please describe the terms and conditions of provenance policy.**

There does not seem to be a formal provenance policy outlining ownership histories, conditions of the artworks at the time of inclusion into the dataset, and legal documentation related to transfers. The artworks were sourced from Wikiart dataset governed by the following terms of use

`https://www.wikiart.org/en/terms-of-use`

**Does the dataset include ownership histories, condition of the artworks at the time of procurement/inclusion into the dataset, and legal documentation related to the transfers? Please describe any available information related to ownership histories.**

Minimal details relating to histories of data instances are captured in WikiArt. For example, metadata associated with Wikiart images contain information about location of the artwork (private collection, museum, etc.), license information, etc.

**What type of information about the music piece/visual artwork's attribution and origination is included? Is it limited to a single creator, or does the dataset afford other categories such as the name of the person who commissioned the artwork, workshop assistants, symphony performers' names, its owners or stewards over time, names of remixers or transliterators, etc.? Please list any available information concerning the people associated with the artwork.**

The information pertaining to artwork's attribution is limited to a single creator, namely, the artist.

**From which source(s) does the dataset draw authenticity verification and title of artworks and music pieces? Please name specific cultural institutions, private or public repositories, and all other sources. Please describe the types of personnel, organizations, and processes involved in the production of authentication and title information.**

ArtEmis is built upon the Wikiart dataset `https://www.wikiart.org` [78].

Artworks in Wikiart are from museums, universities, town halls, and other civic buildings of more than 100 countries.

The original Wikiart dataset allows users to contribute content -

`https://www.wikiart.org/en/How-to-Contribute`

Regarding authenticity and accuracy of the contributed content to Wikiart, it is mentioned that it is the responsibility of the content poster - "You are responsible for the Content that you post to the Service, including its legality, reliability, and appropriateness."

`https://www.wikiart.org/en/terms-of-use`

It is also mentioned that registered users who find discrepancies in the content, can make corrections - points 2 and 3 in `https://www.wikiart.org/en/How-to-Contribute`

**Is information pertaining to the music piece/visual artwork's appreciation included in the dataset? For example, information about multiple interpretations of an artwork/music piece, reviews, labels imprinted on the artworks/music pieces, etc. Please describe.**

WikiArt (and hence ArtEmis) does not include original interpretations of the artworks such as reviews, labels imprinted on the artworks or those by art scholars and connoisseurs.

**Are the values of Indigenous and minoritized groups honored while including the visual artwork/music piece in the dataset? Describe the process or protocol by which visual artworks/music pieces affiliated with these groups have been approved for use in the dataset.**

Indigenous artifacts are underrepresented. The dataset represents a limited selection of art, centered primarily on North American and European fine arts, which excludes other traditions and communities of practice, according to our review of the "art-styles" and "genres" that the ArtEmis developers have incorporated into the filtering tools for exploring the dataset at `https://www.artemisdataset.org/#dataset`.

All the artworks have been sourced from Wikiart dataset. WikiArt presents both public domain and copyright protected artworks. The latter are showcased in accordance with fair use principle: Please see `https://www.wikiart.org/en/about` for details.

### A.3 Data Composition

**What real-world objects or phenomena do the data instances represent (e.g., material objects, performances)?**

The data instances are images of artworks in various media—such as paintings, sculptures, and photographs—although not all artistic media are represented. The ArtEmis developers claim that at

the time of their download from WikiArt in order to build the new dataset, the instances were drawn from 80,031 unique entries representing the work of 1,119 artists. Each data instance includes natural language labels provided by annotators who were sourced by the ArtEmis developers to contribute descriptions of their personal emotional responses when they viewed the image derived from WikiArt.

**If the dataset has been derived from a larger curated corpus, please describe the relationship between the dataset and the larger sample(s) by explaining environmental, sociopolitical, and professional conditions that presently or historically affect the gap between the dataset and total corpus.**

The ArtEmis dataset is built on a download from WikiArt that occurred in 2015. [32, 79] The ArtEmis developers state that, at the time of download, WikiArt contained 81,446 data instances of which they identified 1,415 as exact duplicates. To find and eliminate redundancies, they used the fdupes program [80] and performed manual inspection on pairs of nearest-neighbors artworks by using features of a ResNet-32, pretrained on ImageNet. Afterward, 80,031 instances remained for annotation and analysis. The ArtEmis dataset therefore represents a non-random sample of the current version of Wikiart, which at the time of this writing in 2021 contains some 250,000 data instances. [78]

Under a broader historical lens, the ArtEmis dataset represents multiple layers of curation, operating cumulatively, that resulted in the makeup of WikiArt at the moment in 2015 when it was used as a basis for the new dataset. ArtEmis does not reach beyond WikiArt for sources; hence, it reiterates environmental, sociopolitical, and professional factors that shaped WikiArt as of 2015. Numbering about 80,000 works, WikiArt is predictably a reduction of the total corpus of artistic creation, i.e., all works made globally throughout human history. Both WikiArt and ArtEmis necessarily exclude artworks that were destroyed or lost before the invention of photography. The datasets focus on a limited number of artistic media, styles, historical periods, and geographies. A high-level review suggests that their composition centers on modern Western artistic traditions and preferences, prioritizing certain media and styles from the six most recent centuries and only two continents (North America and Europe). The precise reasons for this reductive result are difficult to document due to the low availability of public information regarding the historical architecture of WikiArt. Generally, a series of choices made by its development team and external contributors determined which artworks are included versus not. In turn, such choices were constrained by prior factors that affected the composition of art repositories, such as those of GLAMs, from which WikiArt derived its data.

**How many instances are there in total, including number of instances per type (e.g., number of artworks/music pieces per artist )? Are there recommended data splits (e.g., training, development/validation, testing) based on the number of instances per type? Please provide a description of these splits, explaining the rationale behind them.**

Wikiart contains 81,446 artworks from 1,119 artists (as downloaded in 2015). The artworks cover 27 art-styles (abstract, baroque, cubism, impressionism, etc.) and 45 genres (cityscape, landscape, portrait, still life, etc.). Details can be found in [79].

**Is the dataset limited by intellectual property restrictions imposed by an external resource (e.g. a cultural institution's decision to exclude works not in the public domain)? Please provide descriptions of external resources and any associated restrictions, including links to access points and policies as relevant.**

WikiArt presents both public domain and copyright protected artworks. Copyright protected artworks are showcased in accordance with fair use principles. Additional details on copyright policy and responses to copyright violations can be found at: `https://www.wikiart.org/en/about`.

The ArtEmis dataset is licensed under the ArtEmis Terms of Use: `https://www.artemisdataset.org/materials/artemis_terms_of_use.txt`

**If data instances are proxies for visual objects, is there information about how the objects were converted to images? E.g., type of camera/model, camera parameters, lighting conditions, etc.? And is there a justification for the choice of the specific viewpoint, camera, and lighting conditions? What aspects of the artwork/object may be missing as a consequence of the specific viewpoint/lighting conditions? Please describe.**

No details about the choices of image capture are provided.

**If data instances are proxies for musical pieces, is information available about how performances were recorded? E.g. type of equipment, studio or live audience setting? Which aspects of the performance may be missing as a consequence of recording conditions? Please describe.**

N/A

**If data instances are proxies for visual objects or fashion pieces, which view(s) of the original objects do data instances represent? E.g., top view, side view, etc.?**

Frontal view. Images are available at multiple resolutions.

**"Does the dataset contain opinions, sentiments, or beliefs that relate to specific social or cultural groups (e.g., religious beliefs, political opinions, etc.) or visualizations/musical renderings that might elicit feelings of anxiety or distress (e.g., those depicting wars, oppression, etc.) ? If so, please describe.**

The original Wikiart does contain sensitive information such as those concerning religious beliefs and political opinions, and artworks that could potentially trigger anxiety such as scenes from wars. ArtEmis, also includes artworks that contain such sensitive information, and those that cause anxiety. For example, see artworks corresponding to categories labeled as "disgust", and "fear" in Figure 6 of the paper.

### A.4   Data Collection

**How were the individual visual artworks or music pieces collected? Were they obtained directly from museums, galleries, music labels, archives, etc.? Or were they derived from existing ML datasets? Please provide details concerning the data acquisition process**

The individual visual artworks were obtained from the publicly available Wikiart dataset.

The original Wikiart dataset contains about 250000 artworks (as of 2021) by 3000 artists across 8 languages (https://www.wikiart.org/en/about). The artworks in Wikiart are from museums, universities, town halls, private collections, and other civic buildings of more than 100 countries.

The annotations of the artworks in ArtEmis were obtained by AMT workers– at least 5 annotators were asked to annotate each artwork to express their dominant emotional reaction to the visual artwork along with an utterance explaining the reason behind their response.

**Is there any institutional (e.g., museum, music label, etc.) policy governing the data collection process? Please describe any applicable institutional ethical standards/rules followed in collecting the data.**

Not explicitly mentioned

**Are all relevant stakeholders (e.g. artists, auction houses, galleries, museums, music labels, composers, legal experts, archivists, etc.) consulted in the collection process? For example, how has the dataset collection process been informed by art conservationists, art historians, and other stakeholders (e.g., Indigenous groups) in order to ensure cultural relevance and longevity? Please describe.**

It seems the authors and the AMT workers are the only stakeholders involved in the data collection process.

**Is there information on the IP rights and cost of acquisition of an artwork or a music piece? Is there information pertaining to their usage, display, reuse, sale, reproduction, and transformation? Please describe all relevant terms and conditions.**

There is no explicit information available about IP rights and cost of acquisition of individual artworks. `https://www.wikiart.org/en/terms-of-use` lists the terms of use of the original Wikiart from which the artworks in ArtEmis were sourced. The terms of use of ArtEmis can be found at

`https://www.artemisdataset.org/materials/artemis_terms_of_use.txt`

**Is there permission to collect/view/reproduce culturally sensitive information? For e.g., those pertaining to Indigenous and minoritized groups, secret objects, sacred objects, etc. that may be forbidden from public view? Please describe the details related to consent request and approval from the concerned individuals and/or communities.**

All the artworks are sourced from Wikiart. According to the copyright terms of Wikiart, copyrighted items are also included , and are showcased under the fair use principles as listed here

`https://www.wikiart.org/en/terms-of-use`

**Do the included works provide wide ranging and multi-dimensional narratives? Please describe how diversity and inclusion has been ensured in the data collection process—for example, demonstrate that the collection is not skewed by providing the number of works across different artists, sources, communities, and geographies; describe how the collection upholds ethical values of the groups it includes and showcases diverse perspectives.**

The ArtEmis dataset contains 81,446 artworks from 1,119 artists, covering artwork created as far back as the 15th century, to modern fine art paintings created in the 21st century. The artworks cover 27 art-styles (abstract, baroque, cubism, impressionism, Ukiyo-e, Rococo, etc.) and 45 genres (cityscape, landscape, portrait, still life, etc.). The dataset seems to be largely a North American-Europe centric dataset exclusive of Indigenous peoples living on those continents and other continents

There are 439,121 explanatory utterances and emotional responses. The resulting corpus contains 36,347 distinct words and it includes the explanations of 6,377 annotators who worked in aggregate 10,220 hours to build it.

The average length of the captions of ArtEmis is 15.8 words (which is significantly longer than the average length of captions of many existing captioning datasets). The authors also provide a linguistic analysis of the captions to demonstrate its "richness and diversity" —by means of listing the average number of pronouns, adjectives, verbs, and adpositions, unique word counts, sentiment analysis, and genre analysis.

Data instances in ArtEmis are accompanied by annotations reflecting subjective judgements and interpretations of the artworks in question, including affective responses and textual explanations of the affective responses. ArtEmis dataset developers recognize the subjective nature of these judgements, and indeed characterize the subjective nature of the judgements as a feature of the datasets. However, no information about the social identities of the annotators is provided in the accompanying dataset documentation so it is difficult to determine whose perspectives are included in the final dataset.

### A.5   Data Pre-processing, cleaning, and labeling

**Were there any specific pre-processing steps (e.g. filtering, resizing, rotations, conversion from color to grayscale, etc.) that might have resulted in loss of information? Please describe**

When displaying the image of an artwork in AMT, the authors scaled down the largest size of the image to 600 pixels, keeping the original aspect-ratio (they did not apply any scaling if the largest size was less than 600 pixels). The authors mention that this "scaling was done to homogenize the presentation of our visual stimuli, and crucially to also reduce the loading and scrolling time required with higher resolution images." Please see `https://www.artemisdataset.org/materials/artemis_supplemental.pdf` [79] for details.

**Were there any translations involved in obtaining the relevant metadata? e.g. translating description of an aboriginal artwork from a native language to English? If so, please describe the diligence process to ensure accurate translations.**

There were no translations involved in obtaining the metadata.

**Does the dataset identify any subpopulation (e.g. by ethnicity, gender expression) by virtue of text inscribed on the artwork or its frame? If so, describe whether such terms are captured in the annotation process and the method of transcription. Describe how such terms are interpreted in their original historical context and how the dataset's users and reviewers can trace the terms.**

The dataset does not consider textual inscriptions associated with the artworks, and therefore does not identify any sub-populations that could have been potentially identified by virtue of such textual metadata.

However, the annotation process involved in curating ArtEmis does identify sub-populations (e.g., by profession, social status, gender, etc.). For example, one annotation reads "The peaceful look of the aristocratic individuals makes you wonder about their lives."

The annotations contained in the dataset are not necessarily backed by their historical context, instead they reflect the subjective opinions of the AMT workers with regards to the emotion triggered in them, looking at the artwork.

**Does the dataset identify any subpopulation in languages other than modern English (e.g. ancient Greek, modern French, modern creole and pidgin languages) or scripts other than standard English characters (e.g., types of Arabic script)? Describe how the dataset documents subpopulations by language(s) and script(s).**

The dataset does not identify sub-populations in languages other than modern English.

**Is there a policy/guideline related to the way labeling/annotation was done? For e.g., any (minimum and maximum) limit on the number of words/characters to describe the artwork, the relevant contexts (cultural, historical, social, etc.) the annotation should cover. Any specific protocol to label based on the type of artwork? Please describe. If the labels were already obtained from existing GLAMs, is there documentation pertaining to how the labels were created? For example, were there any institution specific guidelines? Please describe.**

The average length of the captions of ArtEmis is 15.8 words. On average, each caption contained 4 nouns, 0.9 pronouns, 1.6 adjectives, 3 verbs, and 1.9 adpositions.

With regards to the diversity of captions per image, the number of unique nouns (normalized averages) in an image caption was 3.4, unique pronouns (normalized averages) was 0.6, unique adjectives (normalized averages) was 1.5, unique verbs (normalized averages) was 2.4 , and unique adpositions (normalized averages) was 1.2

To the best of our knowledge, the AMT workers were specifically not instructed to write captions satisfying certain criteria such as minimum word count, parts of speech, etc.

The AMT workers were not instructed to specifically include historical, cultural, social, religious or other such contexts in labeling the artworks. Instead, they were asked to identify the dominant emotion triggered by the artwork ( in their opinion), and also to provide an explanation for the same.

Detailed rater guidelines are provided in the supplementary material [79].

The labels were not obtained from existing GLAMs.

**What information is included in the labels? For example, artist's name, artist's nationality, date associated with the artwork, associated art movement (Byzantine, Renaissance, etc.), genre (classical music, portraits, etc.), art material (oil, paint, etc.), source of the artwork (museum name, name of the recording label, etc.), etc. Please describe.**

Data instances in WikiArt are accompanied by the following pieces of metadata (not every data instance has complete meta-data information):

- Artwork name
- Artist name
- Date
- Style (e.g. Expressionism, Romanticism, Impressionism, etc)
- Genre (e.g. (cityscape, landscape, animal painting, etc)
- Media (e.g. oil, canvas, sculpture, etc.)
- Location (e.g. private collection)
- Dimensions (width x height)
- Copyright (Public Domain or Fair Use)
- Tags (e.g. cat, fruit, glove, etc.)

ArtEmis additionally provides categorical and natural language text annotations for the WikiArt images. Each Data instance is accompanied by 5+ annotations, each of which includes: Dominant

emotional reaction (categorical response, chosen from: anger, disgust, fear, sadness, amusement, awe, contentment, excitement, something-else) Explanation for the emotional response (free-form text response)

**Is there sufficient provenance information (e.g. historical, political, religious, cultural evidence, artist, date, art period, etc.) pertaining to the artwork to generate reliable labels? If not, describe how the labels were obtained/created?**

Some data provenance information (e.g. artist, date, art genre) is associated with data instances as part of the information available with Wikiart. However, no data provenance information was provided to AMT workers during the labelling and captioning task.

**Describe the background of the annotators (e.g., education, geographic location, their socio-cultural contexts, etc.) Do they possess relevant qualifications/knowledge to label the instance? If so, please justify. Are there some aspects of their specific backgrounds (e.g. political affiliations, cultural contexts, etc.) that could have biased the annotations? If so, please describe.**

To the best of our knowledge, there is no documentation concerning the background of annotators.

## A.6   Data use and distribution

**What are the terms and conditions information pertaining to the use and distribution of the dataset? For example, are there any licenses, required clearances from ministries of culture? Please describe the terms and conditions related to such license/clearance.**

The following are the terms and conditions for the use and distribution of the ArtEmis dataset

`https://www.artemisdataset.org/materials/artemis_terms_of_use.txt`

The code is released under MIT license `https://www.artemisdataset.org/materials/MIT_license.txt`

**Is there a link or repository that lists all papers/works that have used the dataset? What are some potential applications and scenarios where the dataset could be used?**

The ArtEmis paper demonstrates the use of the dataset for predicting dominant emotion in visual content and text, and also to generate explanations for the predicted emotions.

The Papers with Code tool lists papers/works that have used this dataset. Papers with Code is maintained independently of the ArtEmis dataset creators and may not capture all the uses of the dataset.

The authors state that the dataset could potentially aid in "deep and nuanced understanding of emotions associated with images."

As stated under point 2 of the terms and conditions above, the authors ("the Universities") mention that they do not make any representations or warranties regarding fitness/use for a particular purpose ( with reference to those who have obtained access to data).

Point 2 in terms and conditions: "The Universities make no representations or warranties regarding the Database, including but not limited to warranties of non-infringement or fitness for a particular purpose."

Point 3 in terms and conditions: "Researcher accepts full responsibility for his or her use of the Database and shall defend and indemnify the Universities, including their employees, Trustees, officers and agents, against any and all claims arising from Researcher's use of the Database, and Researcher's use of any copies of copyrighted 2D artworks originally uploaded to `http://www.wikiart.org` that the Researchers may use in connection with the Database."

**If the data concerns people (such as Indigenous groups), have they provided consent for the use and distribution of the dataset in the context of the specific application? If the dataset is going to be distributed, please describe the diligence process for ensuring that there are no existing restrictions on sharing the artworks and metadata.**

There does not seem to be any explicit information about consent request and approval from concerned individuals and/or communities (associated with the artworks). All the artworks have been sourced

from Wikiart dataset. WikiArt presents both public domain and copyright protected artworks. The latter are showcased in accordance with fair use principle:

- as historically significant artworks

- as used for informational and educational purposes

- as readily available on the internet

- as low resolution copies unsuitable for commercial use

Please see `https://www.wikiart.org/en/about` for details.

**Are there any aspects of human rights and dignity that are prone to be suppressed in a specific use or distribution of the dataset? Please describe applications and scenarios where the dataset should not be used or distributed.**

The authors do not explicitly describe scenarios where the dataset should not be used. They mention it is the responsibility of the users to ensure fair use of the dataset ( see points 2 and 3 in ToC)

**If the dataset will be distributed under a copyright or other intellectual property license, and/or under applicable terms of use, please describe whose (individual or group's) interest/stake in the data is likely to be promoted, and whose interest/stake is likely to be suppressed, and why so. Please justify the use/distribution of the dataset given this context.**

Several groups of stakeholders are affected by the authors' choice to make the dataset and code available for distribution under terms of use and a MIT license. Each adoption of the dataset by new parties introduces greater distance between the end-users and people attached to the artworks' creation, cultural significance, and care. The artworks' communities of origin—including artists, audiences, and supporters—are not ensured of continual connection as named individuals or groups with the data. Instead, human associations with the artworks are permitted to be stripped away by new users' design and development decisions about which labels to retain versus discard. The dataset distribution presents further opportunity to loosen ties between the artworks and stakeholders by inserting optionality into whether information is kept for all data instances about their histories with specific art collectors and owners, groups associated with the artworks' reception and location, and caretakers (such as curators and conservators), to name a few examples. In such scenarios, human connections with artworks are potentially forfeited, a foreseeable risk inherent in the authors' decision to share the dat Artaset while releasing considerable responsibility for how the data and metadata will be used, interpreted, and distributed again. By contrast, the authors of thet and their institutions stand to benefit through the mechanism of citation, whether in scholarly literature or broader channels.

### A.7   Data Generation

**Is there any generated content in the dataset? If so, please explain the source data that is used as examples to generate the desired output.**

N/A

**If there are generated contents in the dataset, is there any data licence required for using or distributing the content?**

N/A

**What are the policies or data licences that should be applied if data samples from this dataset are used to create any synthetic / generated data (e.g: as a generated artwork)?**

N/A

**If there are generated contents in the dataset, is there documentation describing the computational costs involved in creating the data? Please describe**

N/A

**What could be some potential biases associated with the generated data? Please describe**

N/A

### A.8   Data Maintenance

**Who is maintaining the dataset and how can they be contacted?**

The dataset is maintained by researchers Panos Achlioptas, Maks Ovsjanikov, Kilichbek Haydarov, Mohamed Elhoseiny, and Leonidas Guibas of Stanford University, King Abdullah University of Science and Technology (KAUST), and Ecole Polytechnique (the "Universities")

The authors can be contacted at *artemis.dataset@gmail.com*

**Is there a policy outlining the dataset update process? For example, will the dataset be updated as and when there is a change in information with respect to a data instance or will the dataset be updated at regular time intervals to reflect all changes in that time interval? Is there a description of the specific changes made- e.g., provenance related, reason for removal of a data instance, or addition of a data instance, etc. Please describe.**

The dataset was recently curated; currently there is no information regarding dataset update.

**Is there is a provision for others to contribute to the dataset? If yes, describe the process for ensuring the authenticity of the information associated with the contributed data instances.**

It looks like others cannot contribute to the ArtEmis dataset at this time.

## B   Case Study: Artsheets for JS Fake Chorales dataset

### B.1   Motivation

**Who funded the curation of the ML dataset, and for what purpose? Who was involved in the curation process? Please list all individuals and/or institutions.**

No information about funding is provided. The dataset was created to address the lack of high quality datasets for learning based modeling of polyphonic symbolic music that also contain human responses to given music samples.

The authors mention that such a generated dataset helps in symbolic music modeling tasks while also improving validation set loss on existing datasets such as JSB Chorales dataset. Further, the dataset creators mention that information concerning human responses to music samples included as part of the dataset can aid in development of objective metrics for measuring qualitative success of generative modeling.

The person involved in the curation is Omar A. Peracha of Humtap, Inc. London, U.K.

**If the ML dataset was sourced from an underlying curated source (e.g. a museum), please outline the factors governing the inclusion of artworks into the source collection.**

344 chorales of music composer Johann Sebastian Bach were used in training the algorithm to create the generated music samples. The Bach chorales were sourced from music21 toolkit [81]. A number of institutions and individuals have contributed content to music21. Please refer to `https://web.mit.edu/music21/doc/about/about.html` for the list of contributors. We do not have information about the factors governing the inclusion of music pieces into the music21 toolkit.

**If the ML dataset was sourced from an underlying curated source, please outline the factors motivating the choice of the data source.**

There is no specific information pertaining to the choice of the data source other than the authors mentioning about the availability of Bach chorales on this specific source — "To create the JS Fake Chorales dataset, we first develop a learning-based sequence modelling algorithm, KS-Chorus, and train it on the Bach chorales as made available by the music21 toolkit."

### B.2   Data Provenance

**Is there a provenance policy outlining the ethics and conditions of transfer of works in the collection? For example, were the visual artworks or music pieces obtained by means of transfer from other sources? Please describe the terms and conditions of provenance policy.**

The JS Bach chorales have been sourced from the Music21 toolkit [81]. Music21 toolkit contains works of several artists, and these works have been included after obtaining permission from several individuals and groups as listed here.

`https://web.mit.edu/music21/doc/about/about.html`

No specific provenance policy is mentioned as such.

**Does the dataset include ownership histories, condition of the artworks at the time of procurement/inclusion into the dataset, and legal documentation related to the transfers? Please describe any available information related to ownership histories.**

Limited ownership information is provided. The music21 toolkit lists the following with respect to the contributors of Bach chorales — "Margaret Greentree kindly gave permission for distribution of her edited collection of the Bach chorales in MusicXML format as part of the music21 corpus. Her website contains all these chorales in additional formats. Any discoveries we make regarding these chorales are done in her memory."

`https://web.mit.edu/music21/doc/about/about.html`

**What type of information about the music piece/visual artwork's attribution and origination is included? Is it limited to a single creator, or does the dataset afford other categories such as the name of the person who commissioned the artwork, workshop assistants, symphony performers' names, its owners or stewards over time, names of remixers or transliterators, etc.? Please list any available information concerning the people associated with the artwork.**

The music pieces in JS Fake chorales dataset are imitations of the style of a single music composer, namely, Johann Sebastian Bach.

**From which source(s) does the dataset draw authenticity verification and title of artworks and music pieces? Please name specific cultural institutions, private or public repositories, and all other sources. Please describe the types of personnel, organizations, and processes involved in the production of authentication and title information.**

All the chorales used to train the generative algorithm have been sourced from Music21 toolkit, please see: `https://web.mit.edu/music21/doc/about/index.html`. Individual contributors to the collections contained in Music21 have been listed here: `https://web.mit.edu/music21/doc/about/about.html`

Specific information concerning authenticity of titles is not provided.

**Is information pertaining to the music piece/visual artwork's appreciation included in the dataset? For example, information about multiple interpretations of an artwork/music piece, reviews, labels imprinted on the artworks/music pieces, etc. Please describe.**

The JS Fake chorales dataset does not contain information about multiple interpretations of the music pieces.

**Are the values of Indigenous and minoritized groups honored while including the visual artwork/music piece in the dataset? Describe the process or protocol by which visual artworks/music pieces affiliated with these groups have been approved for use in the dataset.**

The JS Fake chorales dataset does not include works of Indigeneous or minoritized groups.

### B.3    Data Composition

**What real-world objects or phenomena do the data instances represent (e.g., material objects, performances)?**

The JS Fake chorales dataset represents 500 synthetic symbolic polyphonic music samples imitating the style of music composer JS Bach.

Additionally, the authors also provide the dataset pre-sliced into 16th note time steps in "js-fakes-16thSeparated.npz". This is a dictionary with keys "pitches" and "chords". The value of each key is a numpy array of 500 sequences. For "pitches", each sequence is a piece from the JS Fakes, itself a list of timesteps. Each time-step has exactly four numbers; one pitch for each of the SATB voices, if a voice is silent at a given time step, its pitch is -1. For "chords", the

format is the same, but each time step instead has just one value representing a chord encoded as per `https://program.ismir2020.net/poster_2-01.html`. All samples are fixed to 2 bars in duration, and melodies are limited to quaver resolution. The samples consist only of pieces in their entirety, with a median bar duration of 11.375 and maximum of 35.5, and the authors state that the rhythmic resolution can fully capture that seen in the original Bach chorales. The dataset also includes anonymised metadata for each of the 6,810 human responses for the music samples. This includes information about the evaluator's level of music education and which specific pieces were guessed correctly or incorrectly and by which evaluator. Also included is information related to how complex a given sample was to identify (as original or generated) by the human evaluators . This includes information pertaining to how many times the sample was played before submitting a response and how long the participant took to submit once they had heard the sample for the first time.

**If the dataset has been derived from a larger curated corpus, please describe the relationship between the dataset and the larger sample(s) by explaining environmental, sociopolitical, and professional conditions that presently or historically affect the gap between the dataset and total corpus.**

Under a broader historical lens, the JS Fake Chorales dataset reiterates environmental, sociopolitical, and professional factors that shaped Music21 toolkit [81]. A high-level review suggests that the composition of JS Fake Chorales dataset is governed by modern Western music traditions, focused on a single music composer. The precise reasons for this reductive result are difficult to document due to the low availability of public information regarding the historical architecture of Music21 toolkit. Generally, a series of choices made by its development team and external contributors determined which music pieces are included versus not. In turn, such choices were constrained by prior factors that affected the composition of recording labels and other sources, from which Music21 toolkit derived its data.

**How many instances are there in total, including number of instances per type (e.g., number of artworks/music pieces per artist )? Are there recommended data splits (e.g., training, development/validation, testing) based on the number of instances per type? Please provide a description of these splits, explaining the rationale behind them.**

There are 500 music samples generated using the original 344 JS Bach chorales. There are also 50 JSF-Extended chorales available, obtained after augmenting the original training data with generated samples. The authors provide the following information related to the training procedure in the paper: "In particular, the edit distance to every single sample in the training corpus must be greater than $50\%$, measured separately across both the entire piece and across the opening few bars. An edit distance below $50\%$ in either case would invalidate the sample and trigger the algorithm to run once more. Furthermore, all samples in the original dataset were transposed as far as possible in either direction while maintaining a singable range for each voice, and the edit distance was similarly validated to these transpositions."

**Is the dataset limited by intellectual property restrictions imposed by an external resource (e.g. a cultural institution's decision to exclude works not in the public domain)? Please provide descriptions of external resources and any associated restrictions, including links to access points and policies as relevant.**

The original JS Bach compositions have been sourced from Music21 toolkit, governed by the following license

`https://web.mit.edu/music21/doc/about/about.html`

**If data instances are proxies for visual objects, is there information about how the objects were converted to images? E.g., type of camera/model, camera parameters, lighting conditions, etc.? And is there a justification for the choice of the specific viewpoint, camera, and lighting conditions? What aspects of the artwork/object may be missing as a consequence of the specific viewpoint/lighting conditions? Please describe.**

N/A

**If data instances are proxies for musical pieces, is information available about how performances were recorded? E.g. type of equipment, studio or live audience setting? Which aspects of the performance may be missing as a consequence of recording conditions? Please describe.**

Specific information about the recording of original JS Bach compositions is not included.

The data instances in the JS Fake Chorales dataset were generated using a generative algorithm called KS-Chorus. No specific details pertaining to the algorithm is provided : the authors state "We do not detail the specifics of the algorithm, input representation or training process here, and instead leave this to a separate work for the future".

**If data instances are proxies for visual objects, which view(s) of the original objects do data instances represent? E.g., top view, side view, etc.?**

N/A

**Does the dataset contain opinions, sentiments, or beliefs that relate to specific social or cultural groups (e.g., religious beliefs, political opinions, etc.) or visualizations/musical renderings that might cause anxiety (e.g., those depicting wars, oppression, etc.) ? If so, please describe.**

Based on our limited experimentation with the web interface, the generated music renderings did not cause anxiety or hurt any sentiments.

## B.4    Data Collection

**How were the individual visual artworks or music pieces collected? Were they obtained directly from museums, galleries, music labels, archives, etc.? Or were they derived from existing ML datasets? Please provide details concerning the data acquisition process**

The data to train the generative algorithm was obtained from Music21 toolkit [81] which in turn has contributions from several groups and individuals.

`https://web.mit.edu/music21/doc/about/about.html`

The annotations were collected from participants sourced via Social media and Amazon Mechanical Turk (AMT).

**Is there any institutional (e.g., museum, music label, etc.)  policy governing the data collection process? Please describe any applicable institutional ethical standards/rules followed in collecting the data.**

There is no information pertaining to specific institutional guidelines.

**Are all relevant stakeholders (e.g.  artists, auction houses, galleries, museums, music labels, composers, legal experts, archivists, etc.)  consulted in the collection process? For example, how has the dataset collection process been informed by art conservationists, art historians, and other stakeholders (e.g., Indigenous groups) in order to ensure cultural relevance and longevity? Please describe.**

It seems the authors, and participants sourced from MTurk and social media are the only stakeholders involved in the data collection process.

**Is there information on the IP rights and cost of acquisition of an artwork or a music piece? Is there information pertaining to their usage, display, reuse, sale, reproduction, and transformation? Please describe all relevant terms and conditions.**

The original compositions in music21 toolkit were included after seeking permissions from the concerned individuals and/or groups.  There is no specific information available with respect to the reproduction or transformation of the original music pieces. Please refer to `https://github.com/omarperacha/js-fakes/blob/main/LICENSE` for license information related to the use and distribution of the generated samples from JS Fake Chorales dataset.

**Is there permission to collect/view/reproduce culturally sensitive information? For e.g., those pertaining to Indigenous and minoritized groups, secret objects, sacred objects, etc. that may be forbidden from public view? Please describe the details related to consent request and approval from the concerned individuals and/or communities.**

N/A

**Do the included works provide wide ranging and multi-dimensional narratives? Please describe how diversity and inclusion has been ensured in the data collection process—for exam-**

**ple, demonstrate that the collection is not skewed by providing the number of works across different artists, sources, communities, and geographies; describe how the collection upholds ethical values of the groups it includes and showcases diverse perspectives.**

The works imitate the style of German music composer JS Bach, and are hence representative of Western music. Non-western music forms and those pertaining to Indigenous and minoritized groups are not showcased in the JS Fake Chorales dataset.

### B.5 Data Pre-processing, cleaning, and labeling

**Were there any specific pre-processing steps (e.g. filtering, resizing, rotations, conversion from color to grayscale, etc.) that might have resulted in loss of information? Please describe**

Not mentioned. The authors state—"We do not detail the specifics of the algorithm, input representation or training process here, and instead leave this to a separate work for the future".

**Were there any translations involved in obtaining the relevant metadata? e.g. translating description of an aboriginal artwork from a native language to English? If so, please describe the diligence process to ensure accurate translations.**

N/A

**Does the dataset identify any subpopulation (e.g. by ethnicity, gender expression) by virtue of text inscribed on the artwork or its frame? If so, describe whether such terms are captured in the annotation process and the method of transcription. Describe how such terms are interpreted in their original historical context and how the dataset's users and reviewers can trace the terms.**

N/A

**Does the dataset identify any subpopulation in languages other than modern English (e.g. ancient Greek, modern French, modern creole and pidgin languages) or scripts other than standard English characters (e.g., types of Arabic script)? Describe how the dataset documents subpopulations by language(s) and script(s).**

N/A

**Is there a policy/guideline related to the way labeling/annotation was done? For e.g., any (minimum and maximum) limit on the number of words/characters to describe the artwork, the relevant contexts (cultural, historical, social, etc.) the annotation should cover. Any specific protocol to label based on the type of artwork? Please describe. If the labels were already obtained from existing GLAMs, is there documentation pertaining to how the labels were created? For example, were there any institution specific guidelines? Please describe.**

Detailed annotator guidelines can be found in page 5 of the paper [35]. The human annotations are stored as a nested dictionary in metadata/js-fakes-dataset.pkl, see `https://github.com/omarperacha/js-fakes` for details.

**What information is included in the labels? For example, artist's name, artist's nationality, date associated with the artwork, associated art movement (Byzantine, Renaissance, etc.), genre (classical music, portraits, etc.), art material (oil, paint, etc.), source of the artwork (museum name, name of the recording label, etc.), etc. Please describe.**

The annotations include human response to the generated music samples. The human evaluators indicate whether a music piece is an original Bach composition or a generated version. The associated metadata also includes information related to how complex a given sample was to identify by the evaluators. Examples include how many times the sample was played before submitting a response and how long the evaluator took to submit once they had heard the sample for the first time.

**Is there sufficient provenance information (e.g. historical, political, religious, cultural evidence, artist, date, art period, etc.) pertaining to the artwork to generate reliable labels? If not, describe how the labels were obtained/created?**

Participants were first exposed to a randomly chosen original Bach chorale. Participants were informed that the given chorale is a genuine example of Bach's music, and they must play it through to the end in order to advance further. Upon doing so, a second piece would appear, which would

be selected at random from the combined set of all pieces, i.e., generated and original (excluding whichever Bach chorale was used as the reference), with equal probability of any sample in this set being selected.

Other than the above information, it seems specific provenance information related to JS Bach Chorales was not provided to the evaluators.

**Describe the background of the annotators (e.g., education, geographic location, their socio-cultural contexts, etc.) Do they possess relevant qualifications/knowledge to label the instance? If so, please justify. Are there some aspects of their specific backgrounds (e.g. political affiliations, cultural contexts, etc.) that could have biased the annotations? If so, please describe.**

Responses were solicited both organically via social media, and through Amazon MTurk. There is some background information available about the annotators such as their self-reported domain expertise level captured as "skill" on a scale of 0-5 as follows.

> 0 - No musical education.
>
> 1 - Some musical education, but no formal qualifications.
>
> 2 - High School level qualifications directly related to music.
>
> 3 - Undergraduate degree directly related to music.
>
> 4 - Postgraduate degree directly related to music.
>
> 5 - Postgraduate degree specialising in the music of J.S. Bach

But, there is no information concerning the social identities of these annotators. Please refer here for details: `https://github.com/omarperacha/js-fakes`

## B.6  Data use and distribution

**What are the terms and conditions information pertaining to the use and distribution of the dataset? For example, are there any licenses, required clearances from ministries of culture? Please describe the terms and conditions related to such license/clearance.**

The dataset is governed by the Creative Commons Attribution 4.0 license. Please refer to `https://github.com/omarperacha/js-fakes/blob/main/LICENSE` for detailed information.

**Is there a link or repository that lists all papers/works that have used the dataset? What are some potential applications and scenarios where the dataset could be used?**

The dataset has been recently released (August 2021). The authors do not yet provide a link/repository that lists works that have used the dataset. Papers with Code webpage can be referred for some information in this regard. Papers with Code is maintained independently of the JS Fake Chorales dataset creators and may not capture all the uses of the dataset.

The authors mention that their dataset could be used for developing objective metrics for assessing the qualitative success of generative models apart from developing models for symbolic music generation.

**If the data concerns people (such as Indigenous groups), have they provided consent for the use and distribution of the dataset in the context of the specific application? If the dataset is going to be distributed, please describe the diligence process for ensuring that there are no existing restrictions on sharing the artworks and metadata.**

The authors mention that no user data of any kind is captured from their web interface, besides the generated sample itself. The webpage `https://omarperacha.github.io/make-js-fake/` does not explicitly mention anything related to user consent. Please refer to `https://github.com/omarperacha/js-fakes/blob/main/LICENSE` for license information related to the use and distribution of the dataset.

**Are there any aspects of human rights and dignity that are prone to be suppressed in a specific use or distribution of the dataset? Please describe applications and scenarios where the dataset should not be used or distributed.**

The authors do not explicitly describe scenarios where the dataset should not be used.

**If the dataset will be distributed under a copyright or other intellectual property license, and/or under applicable terms of use, please describe whose (individual or group's) interest/stake in the data is likely to be promoted, and whose interest/stake is likely to be suppressed, and why so. Please justify the use/distribution of the dataset given this context.**

While the dataset might address the need for high quality generated datasets, it is to be noted that each generated piece introduces greater distance between the end-users and people attached to the music pieces' creation, cultural significance, and care. The music piece's communities of origin—including composers, performers, audiences, and supporters—are not ensured of continual connection as named individuals or groups with the data. Furthermore, this might also raise issues related to artist stereotyping and cultural appropriation.

### B.7 Data Generation

**Is there any generated content in the dataset? If so, please explain the source data that is used as examples to generate the desired output.**

Yes. To create the JS Fake Chorales dataset, the authors developed a learning-based sequence modelling algorithm, called KS-Chorus, and trained it on 344 Bach chorales as made available by the music21 toolkit [81].

Additionally, the authors also use their generated JS Fake Chorales as training data or auxiliary data, there are 50 JSF-Extended chorales available.

The authors mention that KS-Chorus is a generative algorithm for polyphonic music of any number of instruments, where each instrument is itself monophonic. Bach's 4-part chorales are an example of such music.

**If there are generated contents in the dataset, is there any data licence required for using or distributing the content?**

The authors mention that a web application (`https://omarperacha.github.io/make-js-fake/`) is freely available for anyone to sample new chorales with KS-Chorus. The application consists of a single page, including text and a button to trigger sample generation. Once the sampling process is complete, the button is replaced with a MIDI player where the user can preview the newly-generated sample, rendered with a piano synthesiser. The MIDI file will also be automatically downloaded to the user's local hard drive from their browser at that time. The authors state that no user data of any kind is captured from this interface, besides the generated sample itself. Please refer to `https://github.com/omarperacha/js-fakes/blob/main/LICENSE` for license information.

**What are the policies or data licences that should be applied if data samples from this dataset are used to create any synthetic / generated data (e.g: as a generated artwork)?**

The authors have used their generated data for data augmentation (they used their generated JS Fake Chorales as training data or auxiliary data on a common benchmark task, namely modelling the canonical JS Bach Chorales dataset). The authors mention that facilitating data scalability is one of the purposes of their work. Although they do not provide the details of their generative algorithm, they mention that in order to reproduce results from the paper, an existing architecture can be used—`https://github.com/omarperacha/TonicNet`

Please refer to `https://github.com/omarperacha/js-fakes/blob/main/LICENSE` for the license information related to the JS Fake dataset.

Please see `https://github.com/omarperacha/TonicNet/blob/master/LICENSE` for license information related to the use of TonicNet.

**If there are generated contents in the dataset, is there documentation describing the computational costs involved in creating the data? Please describe**

The authors mention that a single composition takes roughly 10 minutes on average to generate when running on a CPU.

**What could be some potential biases associated with the generated data? Please describe**

Artist stereotyping and cultural appropriation could be some issues of concern.

**B.8   Data Maintenance**

**Who is maintaining the dataset and how can they be contacted?**

The dataset is being maintained by Omar Peracha, and the contact information is *omar.peracha@gmail.com*

**Is there a policy outlining the dataset update process? For example, will the dataset be updated as and when there is a change in information with respect to a data instance or will the dataset be updated at regular time intervals to reflect all changes in that time interval? Is there a description of the specific changes made- e.g., provenance related, reason for removal of a data instance, or addition of a data instance, etc. Please describe.**

The authors mention that the dataset will be periodically updated with new unannotated chorales generated by the community with the JS Fake Chorale Generator:

`https://github.com/omarperacha/js-fakes`

The authors state in the paper [35]— "The main JS Fake Chorales repository will be periodically updated to include all samples generated from the web app, and users are informed of this. No user data of any kind is captured from this interface, besides the generated sample itself. While we commit to the persistency of the JS Fake Chorales dataset, and all samples generated from this web app in future included therein, we may in due course choose to discontinue the availability "

**Is there is a provision for others to contribute to the dataset? If yes, describe the process for ensuring the authenticity of the information associated with the contributed data instances.**

As mentioned in `https://github.com/omarperacha/js-fakes`, it seems there is a provision for others to contribute to the dataset. It is to be noted that unannotated chorales generated by the community will be included in the dataset, thus it becomes hard to verify the authenticity of the contributed sources.

