# OpenReview forum: "Artsheets for Art Datasets"
_NeurIPS.cc/2021/Track/Datasets_and_Benchmarks/Round2 — NeurIPS 2021 Datasets and Benchmarks Track (Round 2)_

### Official Review · Reviewer_5cLy · 2021-09-17
**Artsheets for Art Datasets**

**Rating:** 7
**Confidence:** 5
**Correctness:** It is well reasoned and well construc…
**Clarity:** It is clear and well written.

**Strengths:**

The paper is of strong significance to a relatively minor corpus of datasets, specifically those that pertain to artistic works. It is therefore useful for anyone using creative works as inputs to subsequent research or products. The paper seeks to make metadata about provenance, credit, usage limitations, traditional knowledge, and misappropriation generally available, which may provide a moment of reflection that could avoid ethical and social harms.

**Weaknesses:**

The datasheet is presented as a handy and useful way to document a dataset, however this is deceptive. While a data scientist or other practitioner might be tempted to take on the task of filling out this datasheet themselves, to effectively fill out the datasheet offered by the authors, a dataset curator would need significant training and experience in museology, tribal consultation, cultural heritage management, copyright law, and art history. There is no framework for incorporating the necessary professional expertise into the datasheet, and risks giving the impression that anyone can answer the provided questions regardless of their background. While the authors state: "While it may be challenging for the ML community to obtain accurate information concerning data provenance, including domain experts such as art historians and other relevant stakeholders can help overcome some of these challenges and better inform the process of dataset curation," this point should be expanded to include a longer list of experts, and the risks of failing to do so should be discussed.

**Additional Feedback:**

This paper suggests an ambitious undertaking for any dataset curator. Suggest how curators might make tradeoffs between the effort required to fully complete this datasheet and the risks of failing to do so.

**Documentation:**

n/a

**Ethics:**

This risks furthering the toxic meritocracy that charactrizes ML and AI work, giving the impression that practitioners can simulate the diverse expertise needed to compile this datasheet. As stated above under "weaknesses":

While the authors state: "While it may be challenging for the ML community to obtain accurate information concerning data provenance, including domain experts such as art historians and other relevant stakeholders can help overcome some of these challenges and better inform the process of dataset curation," this point should be expanded to include a longer list of experts, and the risks of failing to do so should be discussed.

**Relation To Prior Work:**

Yes, it is clearly related to prior work on datasheets, and makes a needful contribution beyond past work.

**Summary And Contributions:**

The authors offer an update for datasheets to be used for providing metadata about datasets containing artistic works. This builds on prior work like "datasheets for datasets" and shows that that framework is adaptable and updateable for various dataset types.

---

### Official Review · Reviewer_qvDu · 2021-09-20
**A Method for Artistic Dataset Transparency**

**Rating:** 7
**Confidence:** 4

**Strengths:**

One of their proposed strengths is that their protocol focuses "specifically on the unique considerations (such as social, legal, cultural, historical, and environmental factors) that arise in the development of art datasets."

In the creation of artsheets, the team drew upon a wide range of multidisciplinary expertise as well as iterative design methods.

The authors use their protocol in two case studies of existing art datasets.

The authors really researched the artistic domain very well, especially in 3.1 Motivation, 3.2 Data Provenance, 3.3. Data Composition, implicitly demonstrating the importance of domain specific datasheets.

Artsheets includes a mechanism to capture "situated knowledge" of art dataset curators.

One thing that the authors do well is consistently tie their work to outside domains, such as law, which allows us to understand art datasets in terms of complex human systems.


**Weaknesses:**

My main concern with the paper is that I just do not see the need for artsheets from a data ethics perspective motivated in a robust way. There should be more argumentation of why it is extremely important from a data ethics/critical data studies perspective to since this argument is the main angle of the paper in terms of the justification of artsheets. It does not have to be 100% concrete; in fact, there can be a philosophical argument, but there should be a stronger argument than there currently is. I'm rooting for this paper to succeed, and that is one of the things that will really take this paper to the next level.

I think the authors were limited by the page length of NeurIPS. There was so much rich critical theory detail missing. For example, a point would be made in which social science authors would normally be allowed to provide the nuanced arguments for an issue. Another limitation w.r.t. to the limited space was that the authors had to put their art sheet case studies in the appendix. If I'm not mistaken, appendices in conferences are read at the discretion of the reviewer. My concern with the artsheet case study was that it did not necessarily strengthen the paper because they chose datasets that were already well documented and did the artsheet themselves. On one hand, the art datasets seemed to be well documented enough to complete the artsheet, but then it begs the question about how artsheet concretely contributes to the dataset documentation process if some people are already providing the robust information required in artsheet. On the other hand, artsheet is useful for those who need help documenting their art dataset, but if they cannot properly document a dataset then maybe they shouldn't be curating one? I think another thing (which isn't a weakness of this paper, just a point) is the question: why do some people document thoroughly and others not so much?

**Additional Feedback:**

It would be great if the authors could provide the questions in one place for those who want to use artsheets. Extra cool points if the authors put it in a format that is convenient for art dataset curators, for example taking into consideration where the data is housed (e.g., Github).

**Clarity:**

How can artsheets be used by a ghost worker who may not have the extra time in their work schedule to execute the protocol?

What are the implications of a study showing that most museums show white men's art say in terms of the ideals of fairness in relation to the curation of art datasets, especially in a world where the majority of people are not white men? Even if we had artsheets as a way to document artistic dataset curation choices, how would one overcome the potential lack of multicultural representation?

In terms of the ownership question, what are your thoughts on the privacy of private citizens who may have previously owned the art having their names on public display in relation to the art potentially without their knowledge/consent?

I'm glad this question was included: "Are the values of Indigenous and minoritized groups honored while including the visual artwork/music piece in the dataset? Describe the process or protocol by which visual artworks/music pieces affiliated with these groups have been approved for use in the dataset". It's interesting that the issue of stolen artifacts is brought up in section 3.4 Data Collection, but then the authors kind of skirt around it in the artsheet questionnaire. I'm wondering the authors' perspective on museums potentially curating datasets for their stolen Indigenous (used expansively) art work and how you see that relating to artsheets. I think it might be a little harder to hold Western colonial museums accountable since we often know the art works are stolen and they haven't given them back in many years. Also, if this was the case, they could simply lie and say that they honored marginalized people's values in the dataset?

My concern is that while artsheets does a lovely job of creating an overall mechanism to document art dataset curation choices, the authors bring up a lot of background that suggests that even using their artsheet for individual pieces in the dataset may be important, especially if the dataset is diverse (e.g., Nigerian, Chinese, and Chilean pieces in a single dataset). That is, an art sheet for each piece in a dataset might be relevant in the creative arts domain. This is true even from a data ethics perspective in relation to the concerns about Indigenous art pieces that are stolen/misappropriated.

"The ML community should be aware of the cultural nuance associated with the original artworks and music pieces, and must ensure that in using and distributing the dataset, all aspects related to human rights and dignity are upheld." I think this is a good point, but I think there are also HUGE limits to not only cisheterosexual white men in the West who are members of the (petit) bourgeois curating datasets, but also everybody else. For instance, I probably have no business using and distributing a dataset on Hindu art because I know nothing about Hindu art. It is not simply enough for a cishet white man who grew up under capitalism-imperialism to simply "be aware of the cultural nuance", especially when they will explicitly/implicitly reinforce their hegemony in some way when using/distributing the dataset. I hope that makes sense.

In addition, I would have loved a sentence or two about what people were doing with ML for art. The authors kind of say people are using it prolifically (and looking at the references, this is true), but it was unclear as a reader what people are even doing in the domain. The only thing that's memorable is the use of generative models.

**Correctness:**

I would be mindful of your discussion in the introduction of the relationship between data and machine learning since you are submitting to a datasets track that will inevitably attract a critical data studies for ML audience. Particularly, a radical data ethicist might critique the idea that marginalized people can be properly represented in datasets at all, especially when large datasets as the norm tend to violate privacy ethics in terms of failing to ask for consent from those whose data are collected. As you point out, ML for art can result in "cultural appropriation (e.g., cultural misrepresentation, offense, and undervaluing) and representational harm", but what about those people who wish to not be represented at all? Where are the consensual data collection practices come into play? How do we evaluate artistic data that may be okay to collect and that which may not?



**Documentation:**

The authors should provide information on ethical and responsible use of artsheets.

**Ethics:**

If you leave me alone long enough, I can always find some, but none immediately come to mind.

**Relation To Prior Work:**

This work draws heavily upon the canonical "Datasheets for Datasets" by extending it to the art domain.

**Summary And Contributions:**

In Artsheets for Art Datasets, the authors present a method called art datasheets that captures the choices of data curators in creating artistic datasets, citing the importance in relation to the rapid integration of ML in the creative arts domain and the consequent issues with ethical concerns (e.g., cultural appropiation/misrepresentation). In addition, the authors argue that it is important to create standardized dataset accountability/transparency protocols in specific domains in order to appreciate the nuance of areas of expertise. Their main contribution is artsheet questionnaire which extends datasheets to the art domain and two case studies using artsheets. The authors' main audience is presumably dataset curators who should document their choices with artsheets.

---

### Official Review · Reviewer_q4X5 · 2021-09-20
**Artsheets for Art Datasets**

**Rating:** 5
**Confidence:** 4

**Strengths:**

As the authors highlight, this paper addresses an understudied research question. Few papers have looked at providing methods to foster data curation in an ethical way. The contribution is aligned with current literature in the field of data ethics, providing a framework to facilitate data management and promote accountable dataset development.

**Weaknesses:**

The authors' work fits into a very pointed field of research. I am not an expert in the field of art data but from a data ethics perspective the work lacks proper analysis of related work. The only work cited in the field of data ethics is Datasheets for Datasets, which is the basis for building the framework of this paper.

**Additional Feedback:**

The authors state that they are integrating the framework proposed in Datasheets for Datasets, which is interesting, but not sufficient to represent a breakthrough in the field. A literature review on ethical data labeling is missing; for this purpose, some of the most interesting works published on conferences such as FAccT or AIES could have been cited (for example: https://datanutrition.org/, https://doi.org/10.1145/3025453.3026044, https://doi.org/10.1145/3442188.3445940).
To facilitate understanding about the workflow, the authors could have included a summary table of the sections they modified from the reference work (Datasheets for Datasets).


**Clarity:**

The paper is very well written and easy to understand. The authors provide detailed explanations of how the framework works and the process that led to the creation of each section.

**Correctness:**

The evaluation methods sound appropriate to me even though I am not familiar with this specific methodology. The design is appropriate and perfectly aligned with current literature on data ethics and fair machine learning domain.

**Documentation:**

The framework is made available in the Appendix and is sufficiently detailed to support reproducibility. However, as the authors themselves state, the work is still in a preliminary state and thus is not open access.

**Relation To Prior Work:**

The authors discuss how the work differs from a single contribution in the reference literature, which forms the basis for the implementation of their framework.

**Summary And Contributions:**

This manuscript describes an approach to help guide assessment of the ways that dataset design may either perpetuate or shift exclusions found in repositories of art data. In particular, the authors focus on providing a framework consisting of a checklist of questions. The intent is to guide and encourage the data creator to good documentation practice about where the data provenance, why the data set was created, how it was processed, and so on. In addition, the authors provide evidence of how the framework works through two case studies.

---

### Official Review · Reviewer_9rnp · 2021-09-20
**Very good execution, novel topic, but of limited interest and impact to the broad ML community**

**Rating:** 7
**Confidence:** 3
**Clarity:** Yes.

**Strengths:**

The paper studies a rather “niche” topic, which may not be extremely relevant to the broader research community. However, I do find the topic interesting and important.

The paper is very well-executed and covers all the important aspects of art datasets. In particular, it does a great job in tailoring the use of datasheets to art applications.


**Weaknesses:**

Although the topic is interesting and the paper is well-executed, I view the contribution of the current paper as marginal compared to [20]. Taking into account the authors’ comparison in lines 48-62, the current work seems to be an extension and specialization of [20]. While there is great value in creating specific guidelines for art datasets and I agree that this work is important for artists, consumers and institutions, I do not think that this work will be more impactful than [20] for the broader community. What is the additional contribution that the current paper offers to the *broad* ML community compared to [20]?

In general, I got the impression that the paper takes certain design principles and good practices as given. For example, including diverse perspectives is an important guiding principle and I agree that the datasets should be created in such a way. However, it would be better to not take these statements as well-known truths and be transparent about the core principles that drive the selection of the questions in the datasets (and potentially justify these principles with the appropriate evidence from the literature).



**Additional Feedback:**

See my questions and concerns in the previous fields.


**Correctness:**

I would appreciate a more detailed description of the findings from the evaluation of the two datasets,  ArtEmis [29] and JF Fake Chorales [32], in Section 4. The current evaluation seems quite vague, using some general statements such as “both the datasets are largely Western415 centric datasets (a prevailing issue in other ML art datasets as well), not necessarily showcasing diverse perspectives.” It would be great to elaborate in greater detail what issues arose and at which questions (and optionally how they could potentially be addressed) and avoid ambiguous language.

**Documentation:**

Adding links to the datasets (in the references’ list) would be useful.


**Ethics:**

I do not have any particular concerns.


**Relation To Prior Work:**

Yes. See also my comment about [20] (in the weaknesses section).

**Summary And Contributions:**

The paper is motivated by the application of machine learning methods to artistic domains and the harms and misuse that the adoption of such techniques might entail (e.g., cultural appropriation). Many of these issues stem from the inappropriate design and creation of the datasets used. The paper thus designs a checklist, tailored to art datasets, that would guide the construction of the dataset and the decisions of the dataset creator.

Using input from a wide range of experts, the authors tailored the Datasheets [20] questionnaire towards art datasets. The questionnaire includes original Datasheets questions, but also two new question categories, data provenance and data generation as well as questions concerning dataset creator motivation, data composition, collection, pre-processing, cleaning, labeling, use, distribution, and maintenance, all specific to art datasets. The authors illustrate the use of their framework on ArtEmis [29], a visual art dataset and JS Fake Chorales [32], a generated music dataset.

---

### Decision · Program_Chairs · 2021-10-11

**Decision:**

Accept

**Comment:**

The paper presents a new form of datasheets specific to artistic datasets, with the intent of addressing a set of standard questions pertaining to the decisions made by the curators of those datasets. The contribution beyond the seminal "Datasheets for datasets" work is domain specificity--a number of questions are raised that are specific to creative endeavors, e.g., provenance. The datasheet and two example use cases are presented.

The paper is well written and described a topic that is appropriate for the special track, but relatively niche. The set of topics covered by the datasheet is well chosen and thorough, and the work fits into the highly important area of data ethics; the existence of the relevant datasheet may help creators of such datasets to seriously consider some of the ethical implications of their work. In terms of weaknesses, the paper could be better tied into the literature, which would help tie it into the broader machine learning discussion on data ethics. More critically, reviewers are concerned that the expertise necessary to correctly fill in the datasheet is extremely broad, to the point that it may provide dataset curators with a false sense of awareness of individual topics when additional domain expertise is really required.

Overall, reviewers agree that this is a clear paper on an important, if domain limited, topic. The authors' responses to the concerns raised are detailed and thorough.